# Caldera resurgence driven by magma viscosity contrasts

Federico Galetto [1], Valerio Acocella[1] & Luca Caricchi [2]

Calderas are impressive volcanic depressions commonly produced by major eruptions. Equally impressive is the uplift of the caldera floor that may follow, dubbed caldera resurgence, resulting from magma accumulation and accompanied by minor eruptions. Why magma accumulates, driving resurgence instead of feeding large eruptions, is one of the least understood processes in volcanology. Here we use thermal and experimental models to define the conditions promoting resurgence. Thermal modelling suggests that a magma reservoir develops a growing transition zone with relatively low viscosity contrast with respect to any newly injected magma. Experiments show that this viscosity contrast provides a rheological barrier, impeding the propagation through dikes of the new injected magma, which stagnates and promotes resurgence. In explaining resurgence and its related features, we provide the theoretical background to account for the transition from magma eruption to accumulation, which is essential not only to develop resurgence, but also large magma reservoirs.

[1] Dipartimento Scienze Università Roma Tre, L.S.L. Murialdo, 1, Roma 00146, Italy. [2] Department of Earth Sciences, University of Geneva, Rue des Maraîchers 13, 1205 Geneva, Switzerland. Correspondence and requests for materials should be addressed to F.G. (email: federico.galetto@uniroma3.it)

Calderas are subcircular depressions forming during the partial evacuation of magma reservoirs and usually associated with large eruptions. Several calderas are characterised by the uplift, or resurgence, of their sunken floor. This inversion of the negative topography of a caldera consists of the largest and amongst the most protracted types of volcanic deformation, ranging from a few hundreds to a thousand metres and lasting several thousands of years (Table 1). Only the pre-caldera tumescence stage of some magmatic systems may last as long (up to tens of thousands of years), though with lower amount of uplift[1].

While many studies have investigated and explained the causes of caldera collapse[19–21], the processes controlling caldera uplift, or resurgence, have been rarely investigated. Current models commonly relate resurgence to the input of new magma[13,22], possibly stimulated by the pressure drop following a caldera-forming eruption[23]. However, the onset of resurgence, when better constrained, occurs up to a few tens of ka after caldera collapse (Table 1), showing that resurgence does not immediately follow caldera formation. This suggests a temporal decoupling between caldera collapse and resurgence and that the magma intruded responsible for resurgence should not be the residual magma left in the reservoir after collapse; rather, the injection of new magma is expected to drive resurgence. Renewed magma injection may occur either into the former magma reservoir[13,22–25], and/or as shallower sills or laccoliths within the altered intracaldera deposits[7,26–29].

A crucial point is that there has not been any attempt to explain why the newly intruded magma largely stagnates at depth, promoting the resurgence, rather than reaching the surface. In fact, syn-resurgence eruptions, when present, usually cluster outside or along the border of the resurgent area (e.g., Toba, Ischia, Valles and Siwi; Fig. 1) and are commonly sporadic and of small volume. Conversely, the intruded magmatic volumes are usually much higher, as suggested by the volume of the uplifted crust (Table 1), even though this estimate may be affected by a volatile (e.g., bubbles) component, magma compressibility and crustal visco-elastic behaviour[30,31].

Another important feature is that the mechanical energy required to uplift the crust during resurgence is similar to that which may be dissipated during a medium-sized eruption[32]. The fact that magma does not usually erupt in significant volume within the uplifting area during resurgence implies that the available magmatic energy is largely dissipated through uplift, rather than eruptions. All these features support an unexplained abrupt increase in the intrusive/eruptive ratio during resurgence. Thus, the enigmatic and elusive problem remains the understanding of the factors that, after a caldera-forming eruption, despite any increase of fracture density in and around the caldera floor, impede the magma driving the resurgence to erupt.

Notwithstanding the lack of models to explain the causes, some commonalities emerge among resurgent systems, listed below (Table 1). All resurgences are observed within caldera systems experiencing the input of new magma, confirming that resurgence postdates caldera collapse; with the exception of a few medium-sized calderas (e.g., Pantelleria, Ischia; Italy), resurgence is commonly observed in large systems with diameter of tens of km[22,24]; these calderas are usually characterised by the highest amount of degassing, associated with new magmatic input, though non-resurgent calderas may also degas significantly (as Furnas, Azores; Supplementary Table 1, and references therein); with the exception of the mafic Siwi (Vanuatu) and Sierra Negra (Galapagos) calderas, resurgence occurs in felsic magmatic systems, suggesting a link with magma chemistry[22,24]; resurgence usually starts between very few and tens of ka after caldera collapse, but in several cases this interval is poorly constrained (Table 1); resurgence is accompanied by minor regional extension (Table 1), even though not all calderas experiencing minor regional extension develop resurgence. Therefore, while the only necessary condition is that resurgence postdates the formation of a caldera when new magma is injected in the system, none of the common features mentioned above appears sufficient alone to lead to resurgence.

Here we test the possibility that following a caldera collapse the non-erupted magma left into the magmatic system may act as a rheological barrier preventing the eruption of newly injected magma and promoting uplift. The progressive crystallisation of the magma left within the reservoir may lead to a shift from magma eruption to intrusion. This may significantly affect the evolution of a volcanic system, triggering the uplift associated with caldera resurgence and/or developing large magma reservoirs. To test the hypothesis that a rheological barrier prevents the eruption of most of the newly injected magma and promotes caldera resurgence, we perform thermal modelling to capture the temporal evolution of the physical properties of magma in growing reservoirs. The term magma reservoir is here used in the broadest sense, without implication of configuration, depth or magma distribution. On the base of these results, we design experiments to simulate magma injection with rescaled properties similar to those obtained from thermal modelling.

**Table 1 Main features of resurgences at active calderas**

| Name | Ac (km²) | Ar (km²) | Uplift (m) | Vr (km³) | Vv (km³) | Type | T (ka) | tr (ka) | Composition | Supply | Reg. ext. T (mm yr⁻¹) |
|---|---|---|---|---|---|---|---|---|---|---|---|
| Ischia[2,3] | 46 | ~20 | ~1000 | 9 ± 2 | <1 | B | 22 ± 5 | <33 | T | yes | <2 |
| Pantelleria[4,5] | 42 | 7 | ~350 | 1.5 ± 0.3 | 0.5 ± 0.2 | B | 27 | <18 | T, R | yes | <2 |
| Campi Flegrei[3,6] | ~200 | ~30 | ~250 | 3 ± 0.5 | 4 ± 0.5 | D | <4 | <15 | T | yes | <2 |
| Long Valley[7,8] | 450 | ~80 | >400 | ~15 | (≪100) | D, G | (100) | (80) | R, Da | yes | 0.6 |
| Valles[9,10] | 283 | ~65 | ≥1000 | 32 ± 5 | <5 | D, G | (<40) | 27 ± 27 | R, RD | yes | <4 |
| Yellowstone SC[11,12] | 2500 | 214 | ? | >27 | 0 | D, G | (<84) | (<123 ± 9) | R | yes | <4.3 |
| Yellowstone ML[11,12] | 2500 | 164 | ? | 20 ± 7 | (<600) | D, G | ~440 | | R | yes | <4.3 |
| Toba[13,14] | 2350 | 950 | 700 | 320 ± 30 | 20 ± 5 | D, G | ~40 | 33.7 | R, Da | yes | 0 |
| Siwi[15,16] | 36 | 18 | >250 | 4.5 ± 1 | 0.5 ± 0.3 | D, G | (<18) | (2) | Ba, TA | yes | ≥0 |
| Iwo-Jima[17,18] | 63 | 17 | >120 | >1.5 | 0 | D | (>2) | <0.8 | TA | inferred | <2 |

Ac area of caldera, Ar area of resurgence, Vr volume of resurgence, Vv DRE volume of syn-resurgence volcanism, t time between caldera collapse and resurgence, tr duration of resurgence, Supply evidence of new magma supply, Reg. ext. rates of regional extension, B block, D dome, G graben, T trachyte, R rhyolite, Da Dacite, RD rhyodacite, Ba basalt; TA trachyandesite
Values in () are poorly constrained and may not be representative

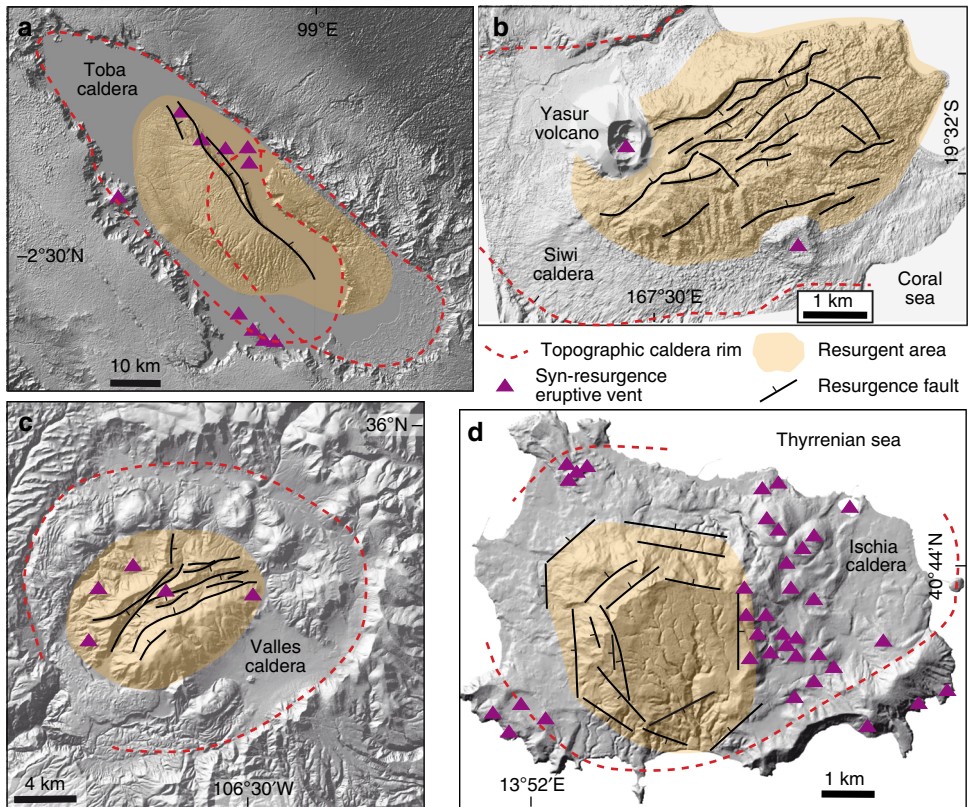

**Fig. 1** Examples of resurgent calderas in nature. Representative resurgences at Toba (**a** Sumatra, Indonesia), Siwi (**b** Tanna, Vanuatu), Valles (**c** New Mexico) and Ischia (**d** Italy), highlighting the outline of the caldera, the extent of the resurgence and the location of syn-resurgence vents (including the inferred ones; see Table 1 for references)

## Results

**Thermal models**. The temporal evolution of magma viscosity as a function of temperature, residual melt composition, water content and crystal content is calculated combining thermal modelling results and existing rheological models[33–35]. Thermal models are performed considering a spherical magma reservoir with maximum volume of 1000 km³, assembled by continuous injection of andesitic–dacitic, water saturated, magma at 1000 °C, at rates between $10^{-1}$ and $10^{-2}$ km³ yr$^{-1}$ at depth of 6–8 km (confining pressure of 200 MPa) and geothermal gradient of 30 °C km$^{-1}$ (refs. [34,36]). The distribution of temperature within the magma reservoir is calculated from the onset of magma injection to the moment at which all the magma cools below solidus temperature.

Magma injection is accompanied by cooling at the contact with the wall rock, which develops a crystallisation front, or transition zone, with decreasing crystallinity and viscosity toward the inner portion of the magma reservoir[37]. This rheological gradient develops from the very beginning of the intrusion and continues for any rate of heat advection lower than the rate of heat diffusion into the wall rock. Therefore, the associated transition zone develops in magma bodies of any geometry both during the construction phase (magma injection) and the pauses in magma injection. Hence, a transition zone is present at all times during the construction of magmatic systems of any shape. In shallower intrusions, such as sills and laccoliths embedded within the altered intracaldera tuffs[7,26–29], the rate of magma crystallisation and development of a transition zone will always be faster than those calculated in our models. Within the transition zone, magma viscosity increases more than 10 orders of magnitude (typically with viscosity $\eta$ between $10^4$ and $10^{15}$ Pa s) (Fig. 2a). The difference in viscosity between the inner portion of this

transition zone in the magma reservoir (defined by a viscosity of $10^4$–$10^8$ Pa s) and any newly injected magma in the post-caldera phase, even if mafic (viscosity of $10^1$–$10^4$ Pa s), is limited. These low viscosity contrasts between injected and resident magma inhibit dike propagation through the resident magma[38,39] and therefore the rise and eruption of newly injected magma during the post-caldera period. The inner part of the transition zone represents an impediment to the propagation of dikes through the magma reservoir, even though its effective role in arresting dikes depends upon its thickness. As a consequence, the growth of the transition zone should be quantified to address its capacity to promote dike arrest and thus the transition from eruption to intrusion. Our models show that the transition zone thickens at the highest rates during the early stage of magma reservoir assembly, when the growth of the reservoir exposes the magma to the contact with colder wall rocksvolume ratio is maximum/ volume ratio is maximum (Fig. 2b, c). The transition zone becomes then progressively thicker during the injection, developing to several hundreds of metres after the injection of 1000 km³ of magma (Fig. 2b, c). The rate of thickening increases, as expected, once magma injection comes to an end (Fig. 2b, c). Here the end of magma injection simulates the possibility that, after caldera collapse, the magma reservoir may not be replenished, or not regularly fed, as for example suggested at Ischia and Pantelleria[4,40]. The thickness of the transition zone increases with increasing volume of the reservoir, with decreasing rate of magma injection and with time from the end of magma injection (Fig. 2b, c).

The thicknesses of the transition zones as a function of time reported in Fig. 2 are absolute minimum estimates, for the following reasons. First, the spherical geometry of the model reduces the rate of heat release with respect to geometries with

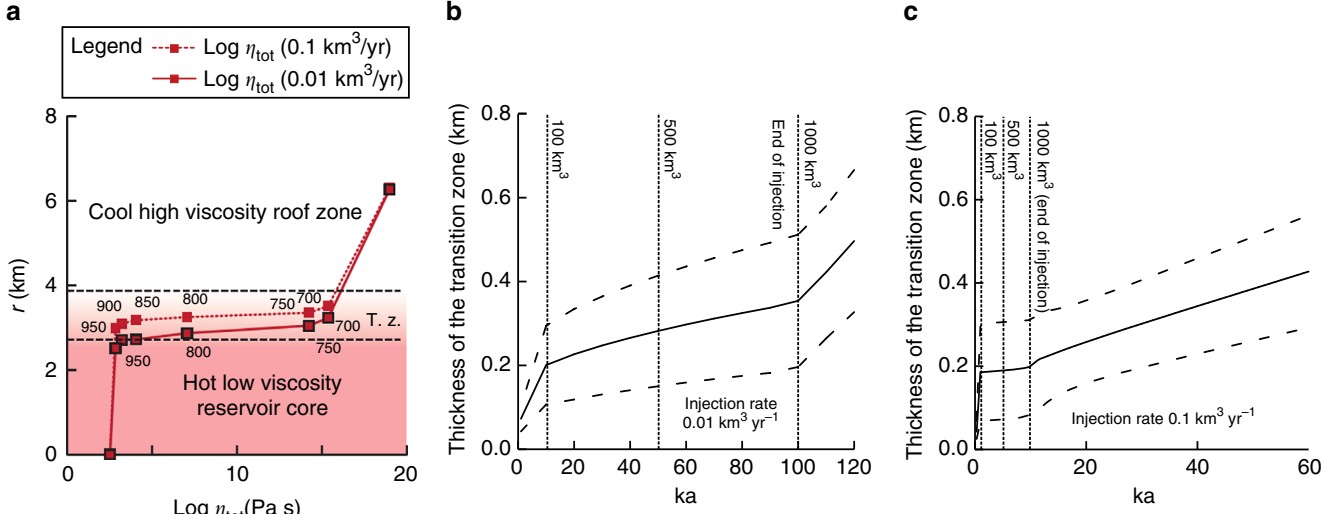

**Fig. 2** Thermal models. Results of thermal models showing the variation of the temperature and magma viscosity ($\eta_{tot}$) along the reservoir thickness. **a** The continuous and dashed lines provide the variation of magma viscosity from the top to the core of the reservoir for magma injection rates of 0.01 and 0.1 km³ yr⁻¹, respectively. The magma viscosity takes into account the effect of crystals on magma viscosity. The numbers next to the lines indicate the temperature in degrees centigrade for the corresponding value of magma viscosity. Black dashed lines represent boundaries of transition zone (T. z.). **b**, **c** Temporal evolution of the thickness of the transition zone (t.z. in **a**) in the magma reservoir for magma injection rates of 0.01 km³ yr⁻¹ (**b**) and 0.1 km³ yr⁻¹ (**c**). Dashed lines refer to the minimum and maximum values of the transition zone (the upper limit considers a magma viscosity range of $10^4$–$10^{15}$ Pa s, while the lower limit is for a magma viscosity range of $10^4$–$10^9$ Pa s)

larger surface to volume ratios: this becomes particularly relevant for tabular reservoirs (sills or laccoliths), especially during the early stages of intrusion. Second, our simulations refer to relatively deep magma reservoirs (6–8 km of depth); as a result, the development of the transition zone in shallower reservoirs will be enhanced (i.e., faster); this also implies that our models may be applied to both magma chambers and shallower intrusions within the intracaldera fill. Third, the heat (i.e., magma) extracted by any eruption is not taken into account. Fourth, relatively high injection temperature with respect to the liquidus temperature of water-saturated andesitic–dacitic magmas is selected here[36]. Fifth, in case of a caldera-forming eruption, evacuation of a significant amount of the eruptible magma (i.e., crystallinity <50 vol.%)[21] may lead to the merging of the remaining non-eruptible magmatic mush, leading to the formation of a single transition zone, approximately two times thicker than the values calculated with our models (Fig. 2). Thus, while a minimum thickness of the inner part of the transition zone resulting from a caldera-forming eruption is of the order of hundreds of metres, a more realistic estimate could be of the order of ~1 km.

We perform additional heat balance calculations to estimate the thermal stability of the transition zone as a function of the magma injection rate (Fig. 3a) and volume (Fig. 3b). For these calculations, we consider the injection of a hot (1200 °C) magma at the base of the transition zone; this temperature was chosen to simulate the injection of mafic magma. The results show that magma supply at injection rates typical of resurgence ($10^{-4}$–$10^{-2}$ km³ yr⁻¹) does not provide sufficient heat to lead to the remelting of a ~500 m thick transition zone (Fig. 3). This thickness may be a minimum estimate, as magma accumulation before large eruptions may last for periods that are longer with respect to those simulated in our thermal models[34,41]. Thus, a thick partially crystallised magma (transition zone) may provide an efficient barrier to inhibit dike propagation and promote magma accumulation. In Fig. 3a, we also report the available surface heat flow and injection rates from resurgences at active calderas. Even though the surface heat flow represents a minimum estimate of the heat released by the associated magma

reservoirs, the resurgences mostly lie on the region of thickening of the transition zone. The only exception is Siwi, whose data are, however, poorly constrained[15]. These data collected for natural system undergoing resurgence show that, thermally, the transition zone can operate as an effective barrier to magma ascent, promoting storage over eruption and leading to resurgence. It is noteworthy that in our calculations we consider only heat lost by conduction from the roof. More plausible scenarios involving, for instance, the extraction of heat by fluids circulating around the magma reservoir would accelerate cooling and further favour the thickening of the transition zone.

**Analogue models.** To test the effectiveness of the transition zone in preventing dike propagation and eruption, we performed experiments using analogue materials. In one set of experiments, vegetable oil (viscosity of $10^{-2}$ Pa s) was injected in a sand pack. In another set, we added a horizontal ~1.5 mm-thick silicone layer (viscosity of $10^4$ Pa s) at different depths within the sand layers to observe the effect of partially crystallised magma (transition zone) on dike propagation. The thickness of the silicone layer was selected to correspond to ~100 m in nature, on the lower bound of the thickness of the inner portion of the transition zone obtained from thermal modelling (Fig. 2). The viscosity ratio between silicone and vegetable oil ($10^6$) was chosen to provide the highest possible contrast of viscosity between resident dacitic magma and injected basalt, therefore offering the least favourable conditions to arrest magma propagation[38]. The vegetable oil was injected at a mean velocity of ~1 m s⁻¹ to prevent any solidification[50] (Supplementary Table 2).

Without the silicone layer, the vegetable oil reaches the surface of the model and erupts, as expected from theoretical calculations showing that dike propagation is favoured by relatively high viscosity contrasts between magma and host rock[38]. Injection and eruption are accompanied by minor vertical (~0.7 mm) and horizontal (~0.7 mm) deformation at the surface (Fig. 4a–d). Conversely, the injection of vegetable oil 1 cm below the silicone layer (the latter at 1.5 cm of depth) leads to doming, the

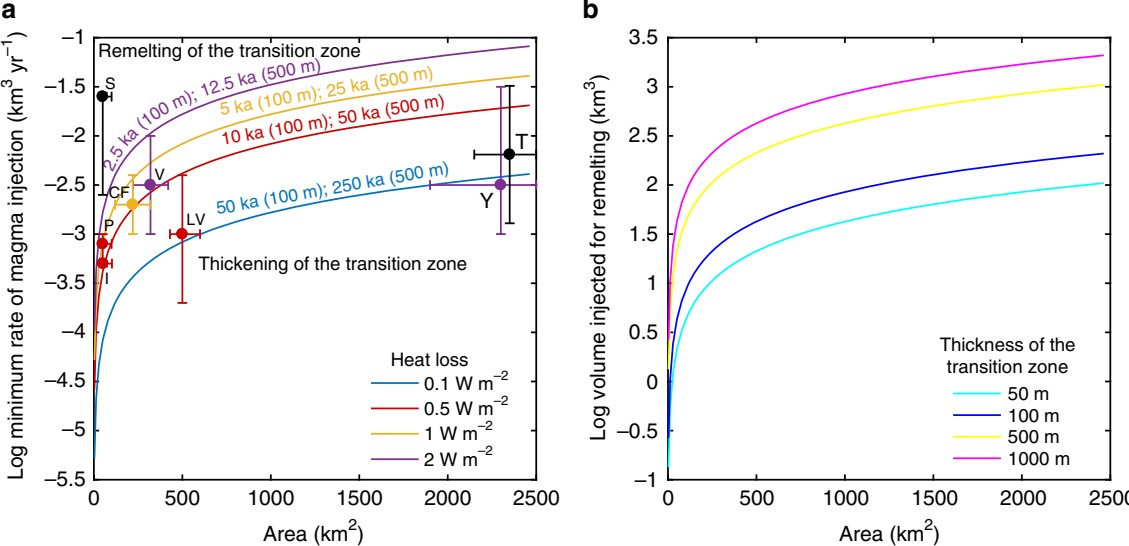

**Fig. 3** Thermal stability of the transition zone. Thermal stability of the transition zone in a replenished magma reservoir. **a** Minimum rate of magma injection required for remelting the transition layer as a function of the roof area of cylindrical reservoirs. The lines show different rates of heat loss per unit area. For lower injection rates, the heat loss is larger than the heat input and the transition layer thickens; for higher injection rates, the transition layer thins. The lines also show the time required for the remelting a transition zone with different thickness (in brackets). Superimposed are the available injection rates and areas of the roof of the magmatic systems from resurgences, with the corresponding errors (which represent the uncertainty of the data); their colours indicate the estimated surface heat flow (black indicates lack of information): I Ischia[42,43], P Pantelleria[4,44], CF Campi Flegrei[45,46], S Siwi[15], V Valles[31,47], LV Long Valley[7,48], Y Yellowstone[11,49], T Toba[13]. **b** Volume of magma required for remelting transition layers of different thickness (colour lines) as a function of the roof area of cylindrical reservoirs

formation of an apical depression bordered by normal faults, and a peripheral subcircular scarp. Eventually, the vegetable oil extrudes along part of this scarp (Fig. 4e–h). Similar behaviour is observed in the experiment with deeper injection (1 cm below the silicone layer, which lies at 6 cm of depth; Fig. 4i–l), in which the oil cannot pierce through the silicone and eventually rises at the edge of the silicone layer. These experiments indicate that the role of the rheological barrier is depth-independent, at least for the simulated depths of few kilometres. Nevertheless, the amount of deformation, as expressed by the uplift at the surface, remains depth-dependent.

These structures develop in all the experiments performed in presence of the silicone layer, and are broadly consistent with previous studies describing or simulating resurgence[51,52] and sill emplacement[53,54], including saucer shaped sills[55]. Importantly, all these structures are also remarkably similar to those observed in resurgences in nature (Valles, Long Valley, Yellowstone, Ischia and Siwi resurgences, Table 1 and Fig. 1).

In the experiments, the silicone layer, having the viscosity suggested by the thermal models, acts as a seal that prevents the vegetable oil from penetrating the sand. This may not exactly represent the real case in nature, where the partly crystallised roof can behave either as a granular slurry or as a coherent interlocking mass of crystals with interstitial liquid, or something in between. Despite these possible variations at the local scale, available field evidence[39] suggests that at the general scale the inner transition zone behaves in a similar way to the silicone in our experiments, largely inhibiting the rise of magma and enhancing its horizontal spreading below, forming sill-like intrusions.

## Discussion

The combination of our thermal and experimental models highlights the importance of a thick transition zone around a reservoir to develop resurgence and its associated structures (Fig. 4e–h). This crucial condition may be met at any depth within a magmatic system, but here we emphasise its occurrence at two main crustal levels, in both cases promoted by caldera formation (Fig. 5). Within the magma reservoir the sudden doubling in the thickness of the transition zone following caldera collapse provides a rheological buffer hindering the ascent of newly injected magma through dikes directly above (Fig. 5b, e). At shallower levels, altered intracaldera tuffs may provide suitable density[56] or stress (because of the lower Young's modulus[57]) barriers to deflect dikes and lead to the emplacement of sills or laccoliths. These tabular intrusions are themselves, together with their upper and lower transition zones, able to induce a wide rheological buffer for the ascent of successive magma, especially during their early stages of growth (Fig. 2a, b). Progressive accumulation under or within these shallow intrusions would drive resurgence (Fig. 5c, f), consistently with geological data[7,26,29]. Magma accumulation may thus develop from rheological barriers at shallow and/or deeper levels in a magmatic system. Accommodation of the volume increase via roof uplift at any depth manifests at the surface as resurgence (Fig. 5).

Therefore, the rheological barrier, whatever its depth, appears the crucial factor to inhibit magma ascent and eruption, favouring magma accumulation and resurgence. Additional processes may contribute, as the intrusion of ring dikes promoting caldera uplift[58] and reactivation of ring faults, but these may have limited importance in producing significant uplift.

As shown in the experiments (Fig. 4) and observed at several resurgences, including Ischia, Pantelleria, Siwi and Toba[2,4,13,16], the new magma-driving resurgence may erupt outside the resurgence and along its borders, through the resurgence faults (Figs. 4 and 5). The magma-producing resurgence should normally not reach the surface within the resurgence area. However, the barrier effect of the transition zone mainly depends upon its thickness. In thinner and more discontinuous transition zones, including those formed both within the magma chamber and within the intracaldera fill fractured during caldera collapse[59],

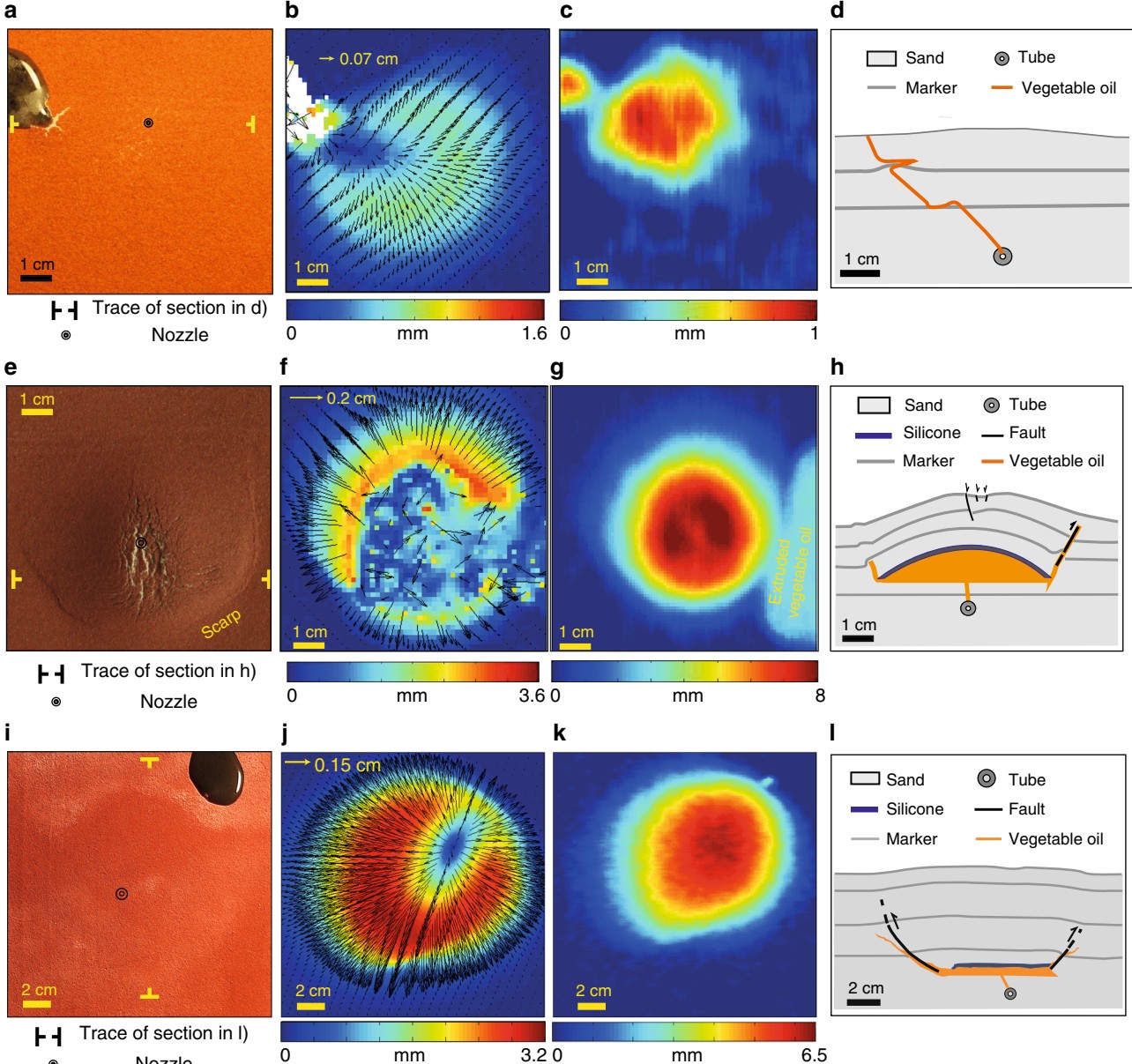

**Fig. 4** Analogue models. **a** Map view of an experiment (RIS 1; Supplementary Table 2) without silicone layer, showing the extrusion of the oil within the sand (*t* = 60 s); a minor deformed area lies to the right of the extrusion point. **b** Cumulative horizontal displacement of the model; colour scale and arrows are proportional to the amount of displacement. **c** Cumulative vertical displacement; colour scale is proportional to the amount of uplift. **d** Section of the experiment (trace in **a**), showing a dike with dip of ~46°, partly deflected along the marker levels. **e** Map view of an experiment (RIS 5; Supplementary Table 2) with shallower (1.5 cm) silicone layer (*t* = 45 s): a dome with apical graben is bordered by a ring scarp. **f** Cumulative horizontal displacement of the model. **g** Cumulative vertical displacement. **h** Section of the experiment (trace in **e**) with a reverse fault, intruded by the oil, at the periphery of a laccolith. **i** Map view of an experiment (RIS 9; Supplementary Table 2) with deeper (6 cm) silicone layer (*t* = 240 s): a dome is accompanied by the extrusion of oil at its upper right edge. **j** Cumulative horizontal displacement of the model. **k** Cumulative vertical displacement. **l** Section of the experiment (trace in **i**) with a reverse fault, intruded by the oil, at the periphery of a sill. Same colours in the scales of the experiments may refer to different deformation values

dikes may propagate and eventually feed small volume eruptions within the resurgence area. This scenario highlights a spectrum of possible outcomes associated with magma injection following caldera collapse, which are linked to the permeability of the transition zone: thicker and non-permeable transition zones mainly lead to resurgence without eruptions within the resurgence area, or non-erupting resurgence; conversely, thinner and more permeable transition zones mainly lead to resurgence accompanied by eruptions within the resurgence area, or erupting resurgence. In this frame, a non-erupting resurgence may experience more uplift than an erupting one. Additional features

may affect these processes: among these, the collapse of a non-coherent block during caldera formation may create subvertical pathways enhancing the eruption of magma during resurgence, within the resurgence area.

Several other features may control the development of the rheological barrier at the top of a reservoir. Among these are the reservoir size, magma supply rate, composition, amount of degassing of the system and tectonic setting. While the transition zone develops in any magmatic system, resurgence focuses in large calderas[22]. Thermal models show that the smallest magma reservoirs (e.g., 30 km³) cool more rapidly. Rapid cooling and

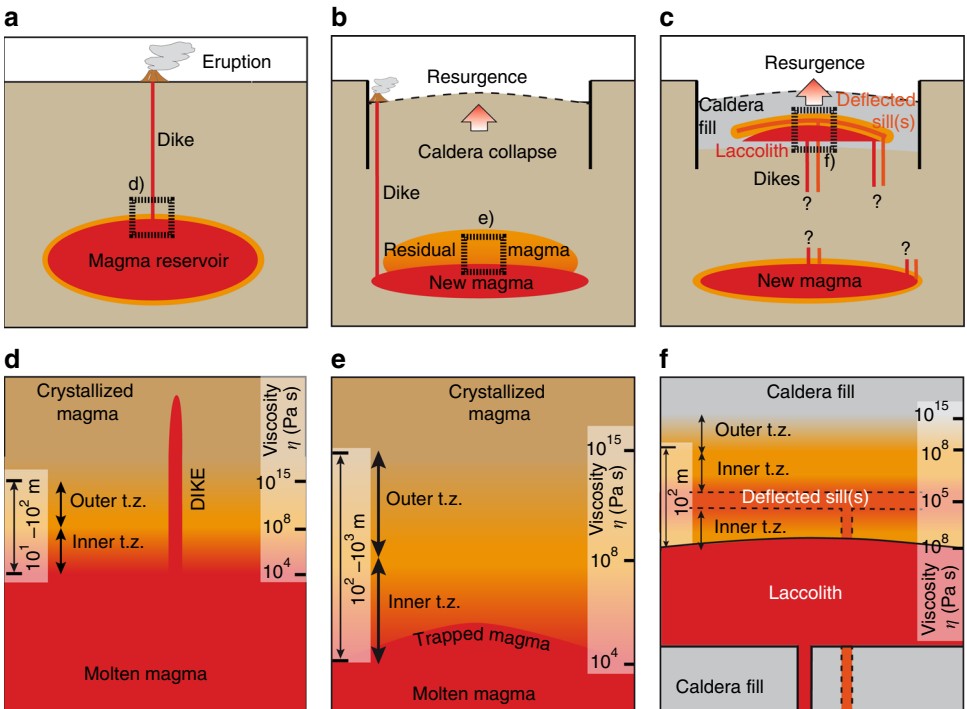

**Fig. 5** Model summarising the conditions for the development of a resurgence after caldera collapse. **a** Conditions developing in a volcano with a thinner transition zone between molten and crystallised magma: dike propagation is promoted and eruptions are more frequent. **b** A caldera-forming eruption partly empties the magma reservoir, leaving a thicker transition zone; this rheological barrier hinders the propagation of new magma through dikes, promoting stagnation at depth and resurgence at the surface; modest eruptions may occur at the periphery of the system. **c** Alternatively, or in combination, to **b**, dikes may arrest at shallower levels, within the physical barrier of the altered intracaldera tuff. Here the arrested dike(s) (orange) may develop one or more sills (orange) within a transition zone (light orange), constituting a barrier for successive dikes (red), stagnating in laccoliths and promoting resurgence; the transition zone related to this latter (red) magma is not shown. The dikes feeding these shallow (orange and red) intrusions may nucleate from the periphery of, and/or within, the magmatic reservoir below, as inferred by the "?" signs. **d**, **e**, **f** illustrate the details of the processes occurring in the dashed boxes in **a**, **b**, **c**, respectively; these detailed boxes are accompanied by probable estimates of the thickness of the transition zone (left) with a given viscosity range (right)

crystallisation to the solidus temperature reduces the time interval over which the viscosity gradient within the transition zone hinders the propagation of newly injected magma. Conversely, in the largest magma reservoirs partially crystallised layer of magma acting as a transition zone can only be removed (i.e., remelted) if very large volumes of magma are injected rapidly (Fig. 3). This condition is not common[4,31,60], explaining the frequent resurgence of the largest calderas. Additionally, the maximum over-pressures achieved in a reservoir by magma injection are inversely proportional to the size of the system[31]. Thus, larger magmatic systems are more difficult to pressurise to the critical levels required to propagate a dike to the surface[31].

The effect of magma supply rate depends upon the size of the reservoir, because relatively low supply rates lead to rapid cooling of the magma reservoir, and therefore to an increase of the viscosity contrast between resident and injected magma that can finally lead to dike propagation, eruption and no resurgence. Very large supply rates, especially in small to medium magma reservoirs fed by high-temperature mafic magma, can cause remelting of the transition zone, its remobilization and lead to eruption and no resurgence. The best conditions for resurgence to occur are the injection of magma at average supply rates in medium to large magma reservoirs. For shallower intrusions within the intra-caldera tuffs, favourable conditions are given by the relatively frequent emplacement of tabular intrusions, so that each successive intrusion becomes emplaced before the previous one has fully crystallised, as also observed at some resurgent calderas[26]. Under these conditions, the transition zone keeps thickening,

maintaining low the viscosity contrast between injected and resident magma and accounting for the resurgence observed at large calderas.

As for the composition, felsic magmas, which are more viscous than mafic ones, require higher critical overpressure to ascent to the surface in dikes[31,38,61]. Additionally, the injection of chemically evolved magma in felsic reservoirs would further inhibit the propagation of dikes to the surface and promote resurgence, because of the minimal viscosity contrast between injected and resident magma. Both these features explain why resurgence is most common in calderas that erupted felsic magmas.

As for the degassing, resurgent calderas usually have high $CO_2$ soil degassing values (Supplementary Table 1, and references therein), associated with well-developed hydrothermal systems. These conditions are particularly promoted by resurgence, where the fluids from the newly emplaced magma escape through a fractured caldera infill and the permeable resurgence faults[23,59]. These features may explain the important degassing observed at resurgent calderas. However, establishing whether and, in case, how degassing may also contribute to resurgence is more challenging and requires additional simulations, not considered here.

Finally, the regional tectonic setting may also control the development of resurgence and the occurrence of syn-resurgence volcanism within the resurgent area. Current resurgences lie in moderately extensional settings (very few mm yr$^{-1}$; Table 1), suggesting that the sustained supply of magma through dikes feeding the tabular intrusions responsible for resurgence may be assisted by some regional extension. This extension, decreasing

**Table 2 Thermal models**

| Parameter | Value |
|---|---|
| $C_{ps}$ (specific heat of crystals) | $1315\ \mathrm{J\,kg^{-1}\,k^{-1}}$ |
| $C_{pm}$ (specific heat of melt) | $1205\ \mathrm{J\,kg^{-1}\,k^{-1}}$ |
| $L$ (latent heat of fusion) | $290{,}000\ \mathrm{J\,kg^{-1}}$ |
| $\rho_c$ (density of crystals) | $2700\ \mathrm{kg\,m^{-3}}$ |
| $\rho_m$ (density of melt) | $2300\ \mathrm{kg\,m^{-3}}$ |
| $T_c$ (critical temperature) | $1073\ \mathrm{K}$ |
| $X$ (crystal fraction) (see Eq. (1)) | $0.5\ (X_c) \leq X \leq 1$ |
| $T_{inj}$ (temperature of injection of the new magma) | $1473\ \mathrm{K}$ |
| $\rho_{inj}$ (density of the new injected magma) | $2900\ \mathrm{kg\,m^{-3}}$ |

Parameters used in Eqs. (2)–(12) (see "Methods" section)

the minimum regional compressive stress σ3, may also enhance the propagation of the newly injected magma via dikes, at least through thinner transition zones, possibly explaining part of the volcanism within the resurgent area. However, a stronger regional extension (several mm yr$^{-1}$ at least) would further promote dike propagation, eventually hindering any resurgence. This condition may occur at the rhyolitic non-resurgent calderas of Taupo Volcanic Zone, New Zealand, experiencing extremely high magmatic productivity, but also significant extension (9–14 mm yr$^{-1}$)[62]. This suggests an optimum in the amount of regional extension (very few mm yr$^{-1}$) in promoting resurgence, beyond which resurgence could be hindered.

The model proposed here to explain resurgence may also have a more general applicability. In fact, stress barriers and/or crustal discontinuities may assist the emplacement of tabular intrusions[57], forming a rheological barrier at any time in the lifetime of a reservoir, also in the pre-caldera stage. The increase in the ratio of intruded versus erupted magma would promote the growth of large reservoirs at depth and tumescence at the surface, eventually leading to catastrophic evacuation through ring dikes at the periphery of the intrusion(s), also controlling the subsidence during the caldera-forming eruption. Therefore, our model may also explain the growth of large reservoirs associated with caldera collapse.

Concluding, the thickening of a rheological barrier within the inner part of the transition zone of a magma reservoir, where the viscosity of partially crystallised magma progressively increases outward, appears a crucial feature for several magmatic and volcanic processes. The transition zone favours the storage of the large amount of magma required for caldera-forming eruptions and inhibits the propagation of dikes after caldera collapse, which is instrumental for resurgence. In particular, at deeper levels, the removal of eruptible magma from a chamber during caldera collapse is an important process contributing to the thickening of the transition zone and resurgence. At shallower depths, the physical state of the altered intracaldera tuff promotes magma arrest and the growth of a tabular transition zone, which constitutes a rheological barrier, promoting the stagnation of successive magma and ultimately resurgence. These barriers develop most prominently in large, felsic and degassing magma reservoirs supplied by average magma fluxes and minor extension (Fig. 1). Our model, describing the shift from eruption-predominant to storage-predominant conditions (Fig. 5), explains the significant increase of the intrusive to extrusive ratio of well-known resurgent calderas. Moreover, our model adequately reproduces all the features (size, composition, degassing and supply rate) of resurgent calderas. The location of syn-resurgence volcanism, usually outside the resurgence or along its periphery, is also explained, as at the reservoir margins magma may not intercept the rheological barrier that impedes dike propagation and can thus erupt. On the

other hand, the syn-resurgence volcanism within the resurgence area may be related to a thinner transition zone, more permeable to magma. Our model provides the theoretical framework for the development of resurgent calderas and may serve to consider long-term strategies to identify the potential location and size of syn-resurgence volcanism in active calderas.

## Methods

**Thermal and viscous models.** Thermal models have been used to calculate the evolution of temperature of a magmatic system, both during its accretion, through injection of new magma, and once magma injection stops.

The models used were previously developed and presented[34] and are based on numerical solutions of the two-dimensional axisymmetric diffusion equation for temperature obtained with the Petrov–Galerkin Finite Element Method on 4-node quadrilaterals. Our calculations include heat advection due to outward displacement of the wall rocks in response to magma emplacement and latent heat generation caused by crystallisation in the magma chamber. All calculations were performed with the following physical parameters: spherical geometries, initial geothermal gradient of 30 °C km$^{-1}$, pressure of 200 MPa (8–10 km depth), heat conductivity of 2.7 W m$^{-1}$ K$^{-1}$, heat capacity of 1 kJ kg$^{-1}$ K$^{-1}$, rock density of 2700 kg m$^{-3}$ and latent heat of fusion of 350 kJ kg$^{-1}$.

The relationships between temperature and magma crystallinity are from a previous experimental study[36], for a composition similar to the Fish Canyon Tuff, whose eruption caused the collapse and resurgence of the La Garita caldera[63]. This composition is appropriate for many resurgent calderas (e.g., Creede, La Pacana, Long Valley, Vilama and Toba)[7,13,63–65]. We used the results obtained at H$_2$O-saturated conditions (i.e., about 6 wt% at 200 MPa)[66].

The crystal fraction ($X$) variation is assumed to vary as a function of temperature ($T$) (Supplementary Fig. 1a) as[34]:

$$X = 1 - \left( \frac{1}{1 + e^{\theta}} \right) \tag{1}$$

Where $\theta = (800 - T)/23$, with the $T$ in °C.

The viscosity of the melt phase ($\eta_m$; Supplementary Table 3) as a function of temperature and H$_2$O content was calculated from a previous model[35]. The effect of crystals on viscosity ($\eta_r$; Supplementary Table 3) was calculated using a previous model[33]. Multiplying the viscosity of the melt ($\eta_m$) by the relative viscosity ($\eta_r$), we obtained the total magma viscosity ($\eta_{tot}$; Supplementary Table 3) as a function of $T$, composition of the melt, crystal and H$_2$O content (Supplementary Fig. 1b).

**Models for the remelting of the transition zone.** The models for the evaluation of the quantity of new magma required for the remelting of the transition zone have been developed using the parameters shown in Table 2.

In order to remelt the transition zone, the heat lost by the magmatic system has to be lower than the heat supplied by magma injection. We considered only the heat that the system loses from its roof by conduction, so that we did not take into account other possible mechanisms of heat dispersion (like interaction with hydrothermal systems) that could significantly enhance the cooling, requiring the injection of even larger volumes of magma for the remelting of the transition zone to take place. We considered that magma with crystallinity >50 vol.% cannot be remobilised[67] (crystallinity of 50% corresponds to a $T = T_c$, see Supplementary Fig. 1, Table 2 and Supplementary Tables 3). Therefore, to remobilise the transition zone, we consider that its crystallinity should drop below 50 vol.%. For the sake of simplicity, we considered 50 vol.% to correspond to 50 wt.% crystals.

The enthalpy content of magma reaching the critical crystallinity (i.e., crystal fraction of 0.5 on a bubble-free base; $X_c$) is:

$$H_c = (C_{ps} \times T_c \times X_c) + [C_{pm} \times T_c \times (1 - X_c)] + L \times (1 - X_c)\ (\mathrm{J\,kg^{-1}}) \tag{2}$$

The enthalpy required to increase the temperature of each 1 kg of magma containing more than 50 wt.% of crystals to $T_c$ is:

$$H_t = H_c - \{(C_{ps} \times T \times X) + [C_{pm} \times T \times (1 - X)] + L \times (1 - X)\},\ (\mathrm{J\,kg^{-1}}) \tag{3}$$

Where $T$ is the temperature between 700 ($T$ solidus) and 800 °C and $X$ the crystallinity at each temperature within this interval (Eq. 1). Assuming that the temperature decreases linearly through the transition zone, the thickness of each portion of transition zone at different temperature can be calculated:

$$\frac{dz}{dT} = \frac{z}{T_{max} - T_{min}}\ (\mathrm{m}) \tag{4}$$

Where $z$ is the thickness of the transition zone and $T_{min}$ and $T_{max}$ are equal to 700 and 800 °C, respectively. The density of the magma within each temperature interval is:

$$\rho_{tot} = \rho_c \times X + \rho_m \times (1 - X)\ (\mathrm{kg\,m^{-3}}) \tag{5}$$

The mass of magma within each temperature interval and unit area is:

$$\text{Mass}_{tr} = \rho_{tot} \times \frac{dz}{dT} \ (\text{kg m}^{-2}) \tag{6}$$

The enthalpy required to remobilise each portion within one $T$ of a column of transition zone of thickness = $z$ and unit area is:

$$H_r = \text{Mass}_{tr} \times H_t \ (\text{J m}^{-2}) \tag{7}$$

Therefore, the total enthalpy required to remobilise the transition zone over the entire area of the magma reservoir is:

$$H_{tot} = \left( \sum_{n=Hr(zmin)}^{Hr(zmax)} H_r \right) \times A \ (\text{J}) \tag{8}$$

Where A is the area of the reservoir.

The parameters of the new magma injected ($T_{inj}$ and $\rho_{inj}$) are reported in Table 2.

We consider that by cooling of 400 °C the injected magma crystallises fully ($x_{inj}$=1) (this is a maximum estimate[68]). The useful enthalpy of 1 kg of the injected magma (i.e., the injected magma can only transfer heat until thermal equilibrium is reached; considering that the transition zone unlocking occurs at 1073 K, only the heat of the injected magma between $T_{inj}$ and 1073 K can be used) is:

$$H = \left( C_{pm} \times T_{inj} + L \right) \\ - \left[ C_{pm} \times T_c \times \left(1 - x_{inj}\right) + C_{ps} \times T_c \times x_{inj} + L \times \left(1 - x_{inj}\right) \right] \ (\text{J kg}^{-1}) \tag{9}$$

The volume of injected magma required for remobilisation (km³) is:

$$V_{inj} = \frac{(H_{tot}/H)}{\left( \rho_{inj} \times 10^9 \right)} \quad (\text{km}^3) \tag{10}$$

Considering that during the injection the resident magma continues to loose heat, the remelting can only happen if the rate of heat input is larger than the rate of heat loss. Assuming a rate of heat loss ($r_{HL}$) (only from the top of the intrusion, as a minimum estimate) and a rate of magma injection, the total heat loss (HL) per year is:

$$HL = r_{HL} \times A \times 365 \times 24 \times 60 \times 60 \quad (\text{J}) \tag{11}$$

The minimum magma flux to compensate the heat loss is:

$$Q_{min} = \frac{(HL/H)}{\left( \rho_{inj} \times 10^9 \right)} \quad (\text{km}^3 \text{ yr}^{-1}) \tag{12}$$

**Analogue models**. The experimental apparatus includes a squared sand box 30 cm long on each side. This box has been filled with a 4 or 8 cm-thick crushed silica powder (mechanical properties in ref. [50]). A nozzle for the vegetable oil injection, with inner diameter of 1 mm, was placed at the centre of the box at a depth of 3, 4 and 7 cm. At depths between 1.5 and 6 cm (Supplementary Table 2), an ~1.5 mm-thick horizontal level of silicone, with variable diameter, has been placed above the nozzle. The vegetable oil was injected using a peristaltic pump.

A camera has been placed above the box to monitor the surface of the model during the experiment. The images have been analysed using the particle image velocimetry technique to detect the horizontal deformation[69]. This is an optical technique that is implemented in the Matpiv software (www.matpiv.com) running under MATLAB®. A laser scanner, placed next to the photo camera above the surface of the experiment, detected the vertical deformation[69].

At the end of each experiment, the models have been covered with 1.5–2 cm of crushed silica powder to preserve the final topography of the experiments. The models were then wet with water and subsequently cut in sections in order to appreciate the subsurface structure. Supplementary Table 2 lists the experiments and all the related parameters.

Model parameters have to be geometrically, kinematically and dynamically scaled, in order to ensure similarities between natural prototypes and experimental results[50]. The main forces to scale are body forces (gravity) and stresses. The dimensionless ratio (Π) between these two forces can be defined as:

$$\Pi = \rho g l / \sigma \tag{13}$$

Where $\rho$ is rock density, $g$ is the acceleration of gravity, $l$ is a linear dimension and $\sigma$ is stress. The cohesion of rock ($c$) has the same dimension of a stress and so we can use the cohesion instead of stress in Eq. (13). As the Π needs to be the same in nature and in the experiments, then $c^* = \rho^* g^* l^*$, which are the ratios between the experiments and in nature for the cohesion, density, gravity and length, respectively. The experiments have been carried out into the terrestrial gravitational field and so the ratio between the $g$ of the models and natural $g$ is 1 ($g^*$=1). The density of the rock of the upper crust is usually between 2500 and 2700 kg m$^{-3}$, while the density of the crushed silica powder used in the experiments is ~1400 (kg m$^{-3}$)[50]. Therefore, the density ratio ($\rho^*$) is ~0.5. Using a length ratio ($l^*$) of $2 \times 10^{-5}$ (1 cm of the model corresponds to ~500 m in nature) the cohesion ratio ($c^*$) needs to be $10^{-5}$. A cohesion of $10^7$ Pa is typical of the rocks that compose the upper crust, which imposes that the cohesion of the experimental sand is ~3 ×

$10^2$ Pa. For this reason, we used crushed silica powder with granulometry between 40 and 200 µm and cohesion of ~300 Pa[50].

The density and viscosity of the vegetable oil depend upon temperature. In Supplementary Table 2, we report the density, viscosity and temperature used in our experiments. The density $\rho$ has been measured using a pycnometer, at known temperature, with the following equation:

$$\rho = m/V \tag{14}$$

Where $m$ and $V$ are the mass and volume of the oil, respectively. The vegetable oil is a Newtonian fluid and its viscosity is independent of the shear stress and strain rates, and is related only to the temperature[70]. The relation between temperature and viscosity of the vegetable oil is described by the Arrhenius' law[70]:

$$\eta = Ae^{(E_a/RT)} \tag{15}$$

Where $A$ is a constant [$(2.3 \pm 0.8) \times 10^{-7}$], $E_a$ is the activation energy ($30380 \pm 850$ J), $R$ is the universal constant of gas and $T$ (K) is the temperature[70]. This equation has been used to define the viscosity of vegetable oil as a function of temperature (Supplementary Table 2). In our experiments, we used a silicone with viscosity of $10^4$ Pa s, in order to keep a viscosity contrast with the vegetable oil of six orders of magnitude (see Supplementary Table 4).

The parameters considered are therefore (Supplementary Table 2): silicone diameter ($D$), depth of silicone level ($d$), depth of vegetable oil injection ($p$), duration of the experiment ($t$), velocity of injection ($v$), density of the vegetable oil ($\rho_{o.v.}$) and sand ($\rho_s$), silicone viscosity ($\eta_{sil}$) and vegetable oil viscosity ($\eta_{o.v.}$), the cohesion of the experimental sand ($c$) and the gravity ($g$).

According to the Buckingham-Π theory, having 11 variables and 3 dimensions, we have to create 8 dimensionless ratios in order to describe the behaviour of our models and test similarities between nature and experiments[52].

The first dimensionless ratio is:

$$\Pi_1 = \frac{\rho_s g D}{c}. \tag{16}$$

The geometrical ratios are represented by:

$$\Pi_2 = \frac{d}{D} \tag{17}$$

$$\Pi_3 = \frac{d}{p} \tag{18}$$

$$\Pi_4 = \frac{p}{D} \tag{19}$$

The density ratio between the vegetable oil and sand is:

$$\Pi_5 = \frac{\rho_{o.v.}}{\rho_s}. \tag{20}$$

The viscosity ratio between the vegetable oil and silicone is:

$$\Pi_6 = \frac{\eta_{o.v.}}{\eta_{sil}}. \tag{21}$$

Two final dimensionless ratios are:

$$\Pi_7 = \frac{vt}{D} \tag{22}$$

$$\Pi_8 = \frac{\rho_{o.v.} p^2}{\eta_{o.v.} t} \tag{23}$$

In Supplementary Table 4, we report the values of these dimensionless ratios for nature and the experiments. The similar values (or same order of magnitude) of the dimensionless parameters, from $\Pi_1$ to $\Pi_7$ in nature and in the experiments highlight an overall appropriate scaling, in terms of used materials and imposed conditions.

However, the dimensionless ratio $\Pi_8$ in nature and experiments differs of two orders of magnitude. In fact, in order to have the same values, the duration of experiments would have been of 3 h, but unfortunately the vegetable oil (which currently provides the best experimental approximation one can get) solidifies in a much shorter time. Despite this discrepancy, we note that in both cases $\Pi_8$ is very small and < 1. This means that both in the experiments and in nature viscous forces exceed inertial forces, as expected[52]. In addition, this dimensionless parameter mainly involves the kinematic behaviour of the intruder, which does not really affect our experimental behaviours and is therefore not discussed in the text.

**Data availability**. The data that support the findings of this study are available from the corresponding author on reasonable request.

# ARTICLE

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

## Acknowledgements

Chris Newhall and Shan de Silva provided helpful insights on degassing and resurgence, respectively. Masashi Nagai provided information on Iwo-Jima. Marco Sacchi provided information on Campi Flegrei. Elodie Brothelande and Marta della Seta provided the DEMs of Siwi and Ischia, respectively. Daniele Trippanera helped in the setting up of the experiments. Guy Simpson wrote the code for thermal modelling. Ben Kennedy and John Stix provided very effective and constructive suggestions. L.C. received funding for this project from the European Research Council (ERC) under the European Union's Horizon 2020 research and innovation programme (Grant agreement No. 677493). V.A. acknowledges the DPC-INGV V2 Project "Campi Flegrei" and the "GeoModAp EEC EV5V-CT94-0464 project". The idea behind this study came from a project on Ischia in 1995, coordinated by Renato Funiciello: without his support we would not have arrived here.

## Author contributions

F.G. run the numerical and analogue experiments; V.A. coordinated the research and wrote the paper; L.C. coordinated the numerical models. All authors contributed ideas and input to the research and writing of the paper.

## Additional information

**Competing interests:** The authors declare no competing financial interests.

