## [Peer Review File · Nature Communications]

Reviewers' comments:

Reviewer #1 (Remarks to the Author):

I enjoyed reading this paper, I think it is well written and presented. The major claim of the paper is that a transition zone around a magma chamber inhibits dyke propagation and promotes uplift following caldera collapse- resurgence. The paper approach is novel with a combined use of thermal modelling to inform magma viscosity contrasts that are in turn used in analogue intrusion experiments with different viscosity fluids. The question of why calderas resurge is as mentioned by the authors of interest to the wider community, and the data presented here is not only of interest in answering this question but is also more generally on the perhaps even more important question of why magma chambers sometimes grow and why they sometimes erupt. The data and approach presented here therefore has a high potential to influence the wider field.

I therefore think that the data could be suitable for publication in Nature Comms. I have highlighted some comments and minor edits in the pdf, that I added whilst reading. However, I would like to raise some important questions about the link between the model results and the observations about resurgence at actual calderas. Unfortunately I don't think the data and model that authors present explain why calderas resurge.

1) The authors data nicely show that during resurgence the eruption to intrusion volume is low. However, my field experience (partially reported in Kennedy et al., 2005, Kennedy et al, 2016) shows that syn resurgence dykes and shallow intrusions maybe commoner than the authors suggest. Indeed, in the recent summary of caldera volcanism Branney and Accocella, 2015 also suggest that when exposed the caldera floor is typically cross cut with dykes. I would argue that this observation is supported by figure 1 in this paper where syn resurgence vents are numerous (presumably each syn resurgence vent is fed by a dyke). So I would argue that resurgence is associated with many small volume eruptions and associated with this are many dyking events. This observation is not mentioned by the authors and is at odds with the results of the thermal models as presented. However, the existence of many dykes during resurgence is probably related to the increased crustal permeability following caldera collapse (e.g. Stix et al., 2003) and does not necessarily preclude the application of the authors model. Perhaps the authors could use this observation- e.g. "despite the increased permeability and reduced critical rock strength for dyke propagation caldera magma chambers inflate rather than erupt".

2) Following on from this the thick transition zones in this paper may be a major reason why bigger eruptions don't occur during resurgence I would also like to point out that before a caldera forming eruption magma accumulates rather than erupts in a manner also predicted by the authors model. I would argue that the authors model related more to the precaldra forming eruption stage than to resurgence- it could be the reason why chamber can grow so big without erupting. For this reason I think that data may be better cast on why do caldera magma chambers sometime erupt and sometimes intrude rather than focussing on resurgence.

3) My reason for suggesting this recasting of the data is because I question the model of resurgence that is presented here -that of refilling of a main caldera magma chamber. My view of a resurging caldera is that dykes (usually utilising subsidence or regional structure) do commonly propagate through the caldera floor and help to form laccoliths at shallow levels. It is the intrusion and inflation these laccoliths that drives resurgence rather than the reinflation of the main chamber. I may be biased in my opinion here but I feel the refilling of a large magma chamber model is rather an outdated idea. The few examples that exist where calderas are sufficiently eroded to reveal uplift relationships with sub caldera magma chambers e.g. Lake city caldera, Colorado (Kennedy et al., 2016), Kumano caldera, Japan (Miura, 1999), Tuscon caldera, Arizona, (Lipman,refs) Turkey creek caldera, Arizona, (du Bray and Pallister, 1999) Snowdon caldera, Wales, (Kokelaar, 1992), Grizzly peak caldera, US (Fridrich et al. 1991) Long valley caldera, california (Hildreth et al 2017) all show that the plutons responsible for resurgence were laccoliths and sills intruded within the caldera fill rather than re-inflating the main chamber. This consistent field observation provides a problem for the model of resurgence presented here. I also think that this view is supported by experimental data .

4) The aspect ratio of the experiments themselves should be put into context with the model of

resurgence presented (refilling of original chamber) and other experiments and observations on sills (Kavanaugh et al. 2016) saucer shaped sills (magee et 2017) laccoliths, resurgent structures (Brothelande et al., 2016). My interpretation is that all these experimental studies support a model of shallow generally < 3km sills or laccolith intrusion driving the style of doming commonly observed at calderas. The scaling in the supplementary material shows that the experiments performed at 1.5cm depth corresponds to a real depth only 750m which is much shallower than the conceptual model of refilling the original magma chamber. Especially if you consider 0.5m-1.5km of caldera fill the original chamber would be at a deeper depths 4km+ ? Again these models provide support for shallow intrusion driving resurgence not refilling of the original magma body. Maybe deeper models are needed to support the proposed reinflation of a chamber with a transition zone

5) I would also recommend a bit more explanation of figure 4, it would really help if the real caldera data were plotted on these figure so the reader could see which calderas do and do not reach conditions for remelting.

In summary, despite my (perhaps biased) misgivings about the very specific application of the experiments and thermal models to resurgence. I think that data and model have important implications to the growth of large magma chambers- and may be better applied to explain how large rhyolitic magma bodies can grow so big without erupting before caldera collapse. It could additionally help explain the location of initial circumferential vents that may evolve into ring fractures suitable for subsidence rather than why calderas resurge.

Reviewer #2 (Remarks to the Author):

The authors propose a model of a partly crystallized upper zone of a magma reservoir sealing the system after a caldera-forming eruption, preventing magma from erupting and causing structural resurgence as new magma from deep levels replenishes the system. Overall, I quite like the model, and I think it is a viable explanation for resurgence. The numerical-experimental approach is solid, in my opinion.

However, I feel that the authors have addressed only part of the story. They need to place their model in a larger context. There are some other equally viable models in the literature, and the authors need to provide a balanced discussion of the relative merits and weaknesses of each. In this regard, I have four comments:

1. The authors suggest that little material is erupted from resurging calderas. This is sometimes true, but sometimes not true. Take the example of Long Valley caldera. At least 100 km³ of aphyric rhyolitic magma was erupted in early postcaldera time. Some of the rhyolite was intruded into the subsided caldera block. Some resurgent uplift occurred, but not that much. With >100 km³ coming out, some subsidence may even have occurred at times. Yellowstone has similarities to Long Valley. So this idea of resurging calderas not erupting is incorrect in some cases. The authors may want to think about a spectrum or continuum of processes, with one endmember being a caldera that resurges a lot without erupting, and another endmember being a caldera that erupts a lot without resurging.

2. The authors should address alternatives to their model. Their transition zone acting as a seal within and at the top of the magma reservoir is the driver for resurgence. But processes acting in the roof block also could drive resurgence, in part or in whole. At Long Valley, McConnell et al. (JVGR 67, 41-60, 1995) and Hildreth (JVGR 335, 1-34, 2017) argue that rhyolite sills in the crustal block cause resurgence. At Rabaul Saunders argues cogently for an intruded ring fault driving central uplift (BV 63, 406-420, 2001). Hence the paper would gain in impact if the authors included a discussion of "reservoir" vs. "crustal" models in driving resurgence.

3. The authors describe the transition zone as one of "partly crystallized magma". However, they do not explain the process or processes by which the zone actually forms. Is this from crystallization of new magma? Of old magma? Is it a residual crystal mush? Or is it some other process that is occurring? In this regard, the authors should discuss the implications of the reservoir roof subsiding during caldera collapse, either partway into the underlying reservoir or all the way to the chamber bottom. When a cold roof suddenly enters the reservoir, how does this affect their transition zone?

4. Figure 4 is a nice figure showing the contrasting behaviour. Likewise, I think the authors' discussion is generally quite good. Some of the points above could be profitably incorporated into this discussion. I would suggest caution with the degassing discussion. It is quite speculative. I would argue that in many cases, a new magma that is forcibly replenishing the system loses massive amounts of gas, whether it be H₂O, CO₂, or both, as it rises and decompresses. The result is extreme overpressure, conditions which are right and ripe for an eruption to occur.

John Stix
6 June 2017

Below we reply (in blue) in detail to the comments.

Reviewer 1

The data and approach presented here therefore has a high potential to influence the wider field. I therefore think that the data could be suitable for publication in Nature

Comms. I have highlighted some comments and minor edits in the pdf, that I added whilst reading. However, I would like to raise some important questions about the link between the model results and the observations about resurgence at actual calderas. Unfortunately I don't think the data and model that authors present explain why calderas resurge.

1) The authors data nicely show that during resurgence the eruption to intrusion volume is low. However, my field experience (partially reported in Kennedy et al., 2005, Kennedy et al, 2016) shows that syn resurgence dykes and shallow intrusions maybe commoner than the authors suggest. Indeed, in the recent summary of caldera volcanism Branney and Accocella, 2015 also suggest that when exposed the caldera floor is typically cross cut with dykes. I would argue that this observation is supported by figure 1 in this paper where syn resurgence vents are numerous (presumably each syn resurgence vent is fed by a dyke). So I would argue that resurgence is associated with many small volume eruptions and associated with this are many dyking events. This observation is not mentioned by the authors and is at odds with the results of the thermal models as presented. However, the existence of many dykes during resurgence is probably related to the increased crustal permeability following caldera collapse (e.g. Stix et al., 2003) and does not necessarily preclude the application of the authors model. Perhaps the authors could use this observation- e.g. "despite the increased permeability and reduced critical rock strength for dyke propagation caldera magma chambers inflate rather than erupt".

First, we have to better clarify that many dike-fed vents may certainly develop during resurgence. But, as also suggested by Fig. 1, these mostly lie outside and along the rim of resurgence; few vents have been observed within the resurgence, during resurgence. This confirms the difficulty of the magma to rise within the resurgence area, but not outside, consistently with our model. This has been better specified now at the beginning of the discussion.

Despite this, we concur with the reviewer that during resurgence dikes may still propagate from the intrusion and reach the surface feeding relatively small eruptions within the resurgence area, as sometimes observed. Therefore, we modified the manuscript to better reflect this point at the beginning of the discussion, also meeting the important first point of Reviewer #2. Indeed, we now propose a spectrum of permeability to new magma, from non-permeable thicker transition zones (resurgences not erupting within the resurgence area) to more permeable thinner ones (resurgences erupting within the resurgence area). However, in this study we would also like to emphasize that the volume of resurgence (and thus of the magma emplaced at depth during resurgence) is commonly much higher than the volume erupted during resurgence (See Table 1). This implies a significant storage of magma in the upper crust during resurgence, despite any dike-fed eruption. Therefore, dikes may still propagate, but not as frequently and/or efficiently as they would without the thermal barrier of the inner transition zone. This has been better clarified in the introduction and at the beginning of the discussion.

2) Following on from this the thick transition zones in this paper may be a major reason why bigger eruptions don't occur during resurgence I would also like to point out that before a caldera forming eruption magma accumulates rather than erupts in a manner also predicted by the authors model. I would argue that the authors model related more to the precaldera forming eruption stage than to resurgence- it could be the reason why chamber can grow so big without erupting. For this reason I think that data may be better cast on why do caldera magma chambers sometime erupt and sometimes intrude rather than focussing on resurgence.

We thank the reviewer for this interesting and useful suggestion, which enlarges the significance of our study to the construction of large magma reservoirs associated with caldera-forming eruptions.

The discussion around the evolution of magmatic systems to a phase of accumulation driven by the progressive growth of a transition zone, which could be essential for the accumulation of large magma volumes in the crust, was not part of the original manuscript. However, we consider this suggestion extremely valid and important and therefore we present it in the abstract, at the end of the introduction and in the discussion sections. In addition, we also better specified at the end of the discussion the possible implications of this process, in explaining the location of initial circumferential vents that may evolve into ring fractures suitable for subsidence, as also suggested at the end of this review (please see last point).

While we agree with the reviewer that the formation of the transition zone during magma injection in the crust is important to stimulate the accumulation of magma, we maintain that after a major eruption the partially molten magma left in the reservoir constitutes a major obstacle for the ascent of magma to the surface, which can be important for resurgence as well.

3) My reason for suggesting this recasting of the data is because I question the model of resurgence that is presented here -that of refilling of a main caldera magma chamber. My view of a resurging caldera is that dykes (usually utilising subsidence or regional structure) do commonly propagate through the caldera floor and help to form laccoliths at shallow levels. It is the intrusion and inflation these laccoliths that drives resurgence rather than the re-inflation of the main chamber. I may be biased in my opinion here but I feel the refilling of a large magma chamber model is rather an outdated idea. The few examples that exist where calderas are sufficiently eroded to reveal uplift relationships with sub caldera magma chambers e.g. Lake city caldera, Colorado (Kennedy et al., 2016), Kumano caldera, Japan (Miura, 1999), Tuscon caldera, Arizona, (Lipman, refs) Turkey creek caldera, Arizona, (du Bray and Pallister, 1999) Snowdon caldera, Wales, (Kokelaar, 1992), Grizzly peak caldera, US (Fridrich et al. 1991) Long valley caldera, California (Hildreth et al 2017) all show that the plutons responsible for resurgence were laccoliths and sills intruded within the caldera fill rather than re-inflating the main chamber. This consistent field observation provides a problem for the model of resurgence presented here. I also think that this view is supported by experimental data.

We thank the Reviewer for pointing out this important feature, which gives us the possibility to propose a more general, and thus stronger, model.

Our view of the pivotal role of a magma reservoir in triggering resurgence was based on classic studies (Smith and Bailey, 1968; Marsh, 1984; Kennedy et al., 2012, for the Valles case; De Silva et al., 2015) and some recent data at active calderas, as at Campi Flegrei (Di Vito et al., 2016, now cited). While this process cannot be excluded, we are now better aware of the feasibility of resurgence triggered by shallower bodies (laccoliths, sills) within the intracaldera fill. Even though these shallow intrusions may not account for the entire resurgent uplift at the surface (Fridrich et al., 1991; McConnell et al., 1995; Hildreth et al., 2017), still suggesting an involvement of the magma reservoir below, their role, as the reviewer points out, cannot be neglected. Therefore, we modified the manuscript throughout (introduction, results and discussion sections) to propose a more general model of broader applicability, which includes also the role of shallower magma intrusions on resurgence.

First, we now use the term “magmatic system” to refer more broadly to the region of the crust affected thermally and mechanically by the presence of magma and circulation of

magmatic fluids. We now also refer to “magma reservoir” (here used without implication of configuration, depth or magma distribution), in distinction to “magma chamber” (which implies a discrete body).

Second, we have now taken into account for the possibility to develop a transitional zone with relatively low viscosity contrast between the new magma and the residual one also at shallower levels. In fact, we now refer to the possibility to develop a shallow intrusion (sill or laccolith) within the caldera fill. The latter mainly consists of altered tuff deposits, with relatively low density and Young’s modulus, providing ideal density (Lister and Kerr, 1991) or stress barriers (Gudmundsson, 2011) to develop sills or laccoliths.

We then consider the role of any of these shallow tabular intrusions on the rise of successive magma batches. In fact, the emplacement of tabular intrusions, heating the surrounding crust, is expected to provide a similar rheological barrier to the successive intruded magma as the one we previously described to occur within a magma reservoir. This possibility is well supported by thermal modelling and presented in the diagrams of Fig. 2. Already during the initial stages of magma injection, cooling at the intrusions boundaries promotes a transition zone with higher gradient (Fig. 2a; Caricchi et al., 2014, Caricchi and Blundy, 2015, Annen et al., 2015). This occurs even in the least favourable conditions for the thickening of the transition zone, as we consider a deeper spherical magma reservoir that dissipates less heat than a shallower sill or laccolith of any aspect ratio. The fact that at eroded calderas resurgence has been related to multiple shallow intrusions (Fridrich et al., 1991; McConnell et al., 1995; Kawakami et al., 2007 - updating Miura, 1999; Kennedy et al., 2015; Hildreth et al., 2017) and, in some cases, each successive intrusion was emplaced before the previous one had fully crystallized (Fridrich et al., 1991) further support an extension of our model also to shallower level intrusions. Additionally, the originally presented analogue models confirm that a shallow transition zone serves as a barrier to magma propagation. To determine whether the depth of the barrier may affect our conclusions, as the reviewer suggested, we performed another experiment simulating a much deeper transition zone; this new experiment confirms that the depth of the transition zone is not important (see following point).

Therefore, we propose a more general model of reservoir growth where our rheological barrier may be related to previously emplaced magma in the original reservoir (as we have suggested) and/or to the shallower (1-2 km) emplacement of sills or laccoliths within the intracaldera fill. Both conditions are promoted, in different ways, by caldera formation and thus specifically apply to the post-caldera stage.

We have now modified the text accordingly at the beginning of the results, discussion, Figure 5 (introducing the rightmost column) and even the key-words.

4) The aspect ratio of the experiments themselves should be put into context with the model of resurgence presented (refilling of original chamber) and other experiments and observations on sills (Kavanaugh et al. 2016) saucer shaped sills (magee et 2017) laccoliths, resurgent structures (Brothelande et al., 2016). My interpretation is that all these experimental studies support a model of shallow generally < 3km sills or laccolith intrusion driving the style of doming commonly observed at calderas. The scaling in the supplementary material shows that the experiments performed at 1.5 cm depth corresponds to a real depth only 750 m which is much shallower than the conceptual model of refilling the original magma chamber. Especially if you consider 0.5m-1.5km of caldera fill the original chamber would be at a deeper depths 4km+ ? Again these models provide support for shallow intrusion driving resurgence not refilling of the original magma body. Maybe deeper models are needed to support the proposed reinflation of a chamber with a transition zone.

This is another excellent point. First of all, as suggested, we performed an additional experiment to test the impact of injection depth on our model (new exp. RIS 9, in Fig. 4i to l and in Table 2 SM). The experiment simulates magma injection at higher depth (7 cm to the injection point, corresponding to ~3.5 km in nature) with respect to the experiments presented in the original manuscript. Despite the lower amount of uplift at the surface with respect to shallower models, the new experiment shows the same behaviour as the previous ones; this is testified by the impossibility for the vegetable oil to pierce through the silicone and eventually rise only at the edge of the silicone layer. This new experiment, now included in Fig. 4, indicates that the role of the rheological barrier is depth independent, at least for the common depths of crustal magma reservoirs.

5) I would also recommend a bit more explanation of figure 4, it would really help if the real caldera data were plotted on these figure so the reader could see which calderas do and do not reach conditions for remelting.

We believe that the Reviewer is here meaning to refer to Fig. 3, rather than Fig. 4. We followed the advice of the Reviewer and added the available real caldera data, with associated uncertainties, to Fig. 3. As a result, we also better described the Figure in the caption and in the text.

In summary, despite my (perhaps biased) misgivings about the very specific application of the experiments and thermal models to resurgence. I think that data and model have important implications to the growth of large magma chambers- and may be better applied to explain how large rhyolitic magma bodies can grow so big without erupting before caldera collapse. It could additionally help explain the location of initial circumferential vents that may evolve into ring fractures suitable for subsidence rather than why calderas resurge.

As specified above (point 2), we incorporated these two important suggestions in the abstract, introduction and discussion sections.

Finally, all the points annotated on the PDF by Reviewer 1 have been considered and improved as suggested. Among these, we particularly considered the possible role of regional extension (also adding a new column in Table 1), describing this in the introduction and discussing in the discussion section.

Reviewer 2 (John Stix)

The authors propose a model of a partly crystallized upper zone of a magma reservoir sealing the system after a caldera-forming eruption, preventing magma from erupting and causing structural resurgence as new magma from deep levels replenishes the system. Overall, I quite like the model, and I think it is a viable explanation for resurgence. The numerical-experimental approach is solid, in my opinion.

However, I feel that the authors have addressed only part of the story. They need to place their model in a larger context. There are some other equally viable models in the literature, and the authors need to provide a balanced discussion of the relative merits and weaknesses of each. In this regard, I have four comments:

1. The authors suggest that little material is erupted from resurging calderas. This is

sometimes true, but sometimes not true. Take the example of Long Valley caldera. At least 100 km³ of aphyric rhyolitic magma was erupted in early postcaldera time. Some of the rhyolite was intruded into the subsided caldera block. Some resurgent uplift occurred, but not that much. With >100 km³ coming out, some subsidence may even have occurred at times. Yellowstone has similarities to Long Valley. So this idea of resurging calderas not erupting is incorrect in some cases. The authors may want to think about a spectrum or continuum of processes, with one endmember being a caldera that resurges a lot without erupting, and another endmember being a caldera that erupts a lot without resurging.

We definitely agree with the reviewer on this spectrum of possibilities, which may reflect different thicknesses of the inner transition zone and thus highlight a continuum from more permeable barriers (associated with less uplifted resurgences, erupting within the resurgence area) to non-permeable ones (associated with more uplifted resurgences, not erupting within the resurgence area). This has been now specified in the first part of the discussion.

2. The authors should address alternatives to their model. Their transition zone acting as a seal within and at the top of the magma reservoir is the driver for resurgence. But processes acting in the roof block also could drive resurgence, in part or in whole. At Long Valley, McConnell et al. (JVGR 67, 41-60, 1995) and Hildreth (JVGR 335, 1-34, 2017) argue that rhyolite sills in the crustal block cause resurgence. At Rabaul Saunders argues cogently for an intruded ring fault driving central uplift (BV 63, 406-420, 2001). Hence the paper would gain in impact if the authors included a discussion of “reservoir” vs. “crustal” models in driving resurgence.

This is another good suggestion, which goes partly (the Long Valley case) along the lines of point 3 of reviewer #1.

Considering Long Valley, we have now taken into account for the possibility to develop a transitional zone with relatively low viscosity contrast between the new magma and the residual one also within the caldera fill. We first refer to the possibility to emplace a shallow intrusion (sill, laccolith) in the intracaldera fill; the latter mainly consists of altered tuff deposits, with relatively low density and Young’s modulus, providing ideal density (Lister and Kerr, 1991) or stress barriers (Gudmundsson, 2011) to develop sills or laccoliths. These shallow tabular intrusions are in turn expected to provide a similar rheological barrier to the successively intruded magma as the one we previously described to occur within a magma reservoir. This possibility is well supported by thermal modelling and presented in the diagrams of Fig. 2 (for details see also reply to point 3 of Reviewer 1).

Therefore, we propose a more general model where our rheological barrier may be related to earlier emplaced magma in the original reservoir (as we previously suggested) and/or to the shallower (1-2 km) emplacement of sills or laccoliths within the caldera fill.

We have now modified accordingly the text at the beginning of the results, discussion and also Figure 5 (introducing the rightmost column).

As for the example of Rabaul, even though the specific observed uplift is much less than that occurring during resurgence, we understand that the process of ring-dike injection may contribute to promote uplift. Therefore, we included also this possibility at the beginning of the discussion.

3. The authors describe the transition zone as one of “partly crystallized magma”. However, they do not explain the process or processes by which the zone actually forms. Is this from crystallization of new magma? Of old magma? Is it a residual crystal mush? Or is it some other process that is occurring? In this regard, the authors should discuss the

implications of the reservoir roof subsiding during caldera collapse, either partway into the underlying reservoir or all the way to the chamber bottom. When a cold roof suddenly enters the reservoir, how does this affect their transition zone?

Below we reply in sequence to all the questions above.

Injection of magma in the crust is inevitably associated with cooling and crystallisation at the contact with the wall rock; this promotes a crystallisation front with decreasing crystallinity and viscosity from the rim of the magma reservoir toward the centre (e.g., Marsh, 2002). The development of this crystallinity and rheological gradient starts from the very beginning of the intrusion and continues for any rate of heat advection lower than the rate of heat diffusion into the wall rock. We define this region as the transition zone. Therefore, the transition zone develops during both the construction (injection of magma) and repose (interruption of injection) of a magmatic system. The formation of the transition zone is now better described at the beginning of the results section.

Based on evidence from resurgent calderas, we have also specified (results section and beginning of discussion) that the transition zone may mainly consist of: residual magma forming a crystal mush on the roof of a magmatic reservoir, and/or by shallower cooling intrusions (sills, laccoliths) embedded within the heated and altered intracaldera tuffs (see also new Fig. 5).

As for the possible role of the cold sunken reservoir roof on the development of the transition zone, this essentially affects older magma forming a crystal mush on the roof of a magmatic reservoir. Here it is expected that the downward translation of the reservoir roof induces the merging of the non-eruptible magma above and below the eruptible (and now, during caldera collapse, erupted) magma promoting a thickened zone of crystal-rich magma. This has been better specified at the end of the presentation of the thermal models.

4. Figure 4 is a nice figure showing the contrasting behaviour. Likewise, I think the authors' discussion is generally quite good. Some of the points above could be profitably incorporated into this discussion. I would suggest caution with the degassing discussion. It is quite speculative. I would argue that in many cases, a new magma that is forcibly replenishing the system loses massive amounts of gas, whether it be H₂O, CO₂, or both, as it rises and decompresses. The result is extreme overpressure, conditions which are right and ripe for an eruption to occur.

Yes, the points above have all been incorporated into the discussion.

As for the degassing problem, we understand the concern of the Reviewer and now removed any interpretation on the role of degassing on resurgence at the end of the discussion and in the key-words.

We thank you for your kind attention.

Faithfully,

Federico Galetto

Corresponding Author

REVIEWERS' COMMENTS:

Reviewer #1 (Remarks to the Author):

I am very happy with how the authors have conscientiously included the ideas and comments of the reviewers. I am also appreciative of the considerable work and extra experiment performed in light of the review process. The manuscript is much improved and broader in its application making it even more suitable for publication in Nature Communications. I have some minor comments on figure 2a that the authors could tweak if they choose, and some minor edits that authors could choose to include.

Figure 2a. This figure still took me a while to get my head around, again, there is a lot going on ! I should have commented more on this in first round of reviews I think you need to help the reader a bit more.

Is viscosity bulk (total) viscosity or melt viscosity ?

Either label crystallinity on top axis, currently the X looks like a point at 0.75, or I would prefer to remove the crystallinity curve completely as it is a component of the total viscosity curve. It really just makes the graph harder to understand. I think most readers are familiar with the role crystallinity plays on viscosity and the addition of sentence in the text could explain this. Something along the lines of "total viscosity is a function of crystallinity and melt viscosity see supplementary table."

Second I would add the vertical info (right side y axis) about core and transition zone onto main graph. At the moment it is a bit weird to have roof labelled on main graph but these on y axis on right. I suggest you could do this by using your red shaded zone better, write "Hot low viscosity reservoir centre" at bottom in this red shaded zone, then have "the transition zone" labelled maybe with an arrow, then label "Cool high viscosity roof zone".

In the figure caption it took me a while to work out "profiles along sections of radius r". Maybe simpler is "rheological gradients..."

I find the isothermal labelling a bit confusing, there maybe a better way to do this but I couldn't work one out.

Line 172 I think it is worth maybe an additional sentence explaining that remelting maybe be possible if the reintruding magma is more mafic (e.g. Kennedy and Stix, 2007 and lots of other people !).

Line 210 I would tweak this sentence. "Similar behaviour is observed in deeper experiments (...), testified..."

Line 215 You could add a caveat that states that deformation style and eventual vent location may be depth dependent (seems to be indicated by the experiment) and consistent with (Acocella et al.)

Line 246 reactivation of ring faults could also be considered here, not just reinjection

Line 255 "also "seems unnecessary

Line 256 I don't think you need "as observed"

Line 307 I am not sure "ad hoc" is the right word here I suggest remove.

Line 805 I would remove "Ordinary"

Well done authors ! Ben Kennedy

Reviewer #2 (Remarks to the Author):

The authors have substantially improved the manuscript. In my opinion it now reads very well. I have four suggestions for further improvement:

1. The manuscript needs a light English edit for clarity.
2. Rewrite the title slightly to: "Why do calderas resurge?"

3. In the experiments, the silicone layer is effectively a seal which prevents the vegetable oil from penetrating the sand. This is not exactly the same situation as in nature, where the partly crystallized roof can behave either as a granular slurry or as a coherent interlocking mass of crystals with interstitial liquid (or something in between). It would be useful if the authors discussed the limitations of their experiments in this regard.

4. The authors ascribe the relative importances of intrusion vs. volcanism in terms of the thickness of the transition zone. This presupposes a coherently subsiding roof which behaves as a piston. If the roof breaks into two or more pieces, however, vertical pathways are established which could allow eruption of magma.

Second round of revisions (replies on 27-9-2017):

Reviewer #1 (Ben Kennedy):

I am very happy with how the authors have conscientiously included the ideas and comments of the reviewers. I am also appreciative of the considerable work and extra experiment performed in light of the review process. The manuscript is much improved and broader in its application making it even more suitable for publication in Nature Communications. I have some minor comments on figure 2a that the authors could tweak if they choose, and some minor edits that authors could choose to include.

Figure 2a. This figure still took me a while to get my head around, again, there is a lot going on ! I should have commented more on this in first round of reviews I think you need to help the reader a bit more.

Is viscosity bulk (total) viscosity or melt viscosity ?

Either label crystallinity on top axis, currently the X looks like a point at 0.75, or I would prefer to remove the crystallinity curve completely as it is a component of the total viscosity curve. It really just makes the graph harder to understand. I think most readers are familiar with the role crystallinity plays on viscosity and the addition of sentence in the text could explain this. Something along the lines of "total viscosity is a function of crystallinity and melt viscosity see supplementary table."

Second I would add the vertical info (right side y axis) about core and transition zone onto main graph. At the moment it is a bit weird to have roof labelled on main graph but these on y axis on right. I suggest you could do this by using your red shaded zone better, write "Hot low viscosity reservoir centre" at bottom in this red shaded zone, then have "the transition zone" labelled maybe with an arrow, then label "Cool high viscosity roof zone".

In the figure caption it took me a while to work out “profiles along sections of radius r ”. Maybe simpler is “rheological gradients...”
I find the isothermal labelling a bit confusing, there maybe a better way to do this but I couldn't work one out.

Thanks for the helpful comments on Fig. 2a. We have now modified this figure following all the suggestions.

Line 172 I think it is worth maybe an additional sentence explaining that remelting maybe be possible if the reintruding magma is more mafic (e.g. Kennedy and Stix, 2007 and lots of other people !).

To re-calculate the heat required for re-melting of the transition zone we used a temperature of magma injection of 1200°C. This temperature was chosen to simulate the injection of a basaltic magma. In the revised version we have clarified this at lines 160-162.

In the discussion we have, however, specified that the injection of more mafic (and hotter) magma could trigger the re-melting of the transition zone (line 280), especially if coupled with high injection rates.

Line 210 I would tweak this sentence. “Similar behaviour is observed in deeper experiments (...), testified...”

This has been done at present line 201-203.

Line 215 You could add a caveat that states that deformation style and eventual vent location may be depth dependent (seems to be indicated by the experiment) and consistent with (Acocella et al.)

This has been specified at lines 205-206.

Line 246 reactivation of ring faults could also be considered here, not just reinjection

This has been considered.

Line 255 “also “seems unnecessary

“Also” has been removed

Line 307 I am not sure “ad hoc” is the right word here I suggest remove.

Removed.

Line 805 I would remove “Ordinary”

Well done authors ! Ben Kennedy

Removed, and thanks for the useful comments.

Reviewer #2:

The authors have substantially improved the manuscript. In my opinion it now reads very well. I have four suggestions for further improvement:

1. The manuscript needs a light English edit for clarity.

The English has been now re-checked.

2. Rewrite the title slightly to: "Why do calderas resurge?"

According with the editor's requests (we cannot use question marks), the new title is now: "Caldera resurgence driven by magma viscosity contrasts".

3. In the experiments, the silicone layer is effectively a seal which prevents the vegetable oil from penetrating the sand. This is not exactly the same situation as in nature, where the partly crystallized roof can behave either as a granular slurry or as a coherent interlocking mass of crystals with interstitial liquid (or something in between). It would be useful if the authors discussed the limitations of their experiments in this regard.

We have added this appropriate point at the end of result section (analogue models subheading) (lines 212-219). This concept has been also briefly considered at the beginning of the discussion (line 249)

4. The authors ascribe the relative importances of intrusion vs. volcanism in terms of the thickness of the transition zone. This presupposes a coherently subsiding roof which behaves as a piston. If the roof breaks into two or more pieces, however, vertical pathways are established which could allow eruption of magma.

We have now incorporated this point in our discussion (Lines 259-261).
Thanks for the useful comments.

Faithfully,

Federico Galetto

Corresponding Author